# Elicitation of Submerged Adventitious Root Cultures of *Stevia rebaudiana* with *Cuscuta reflexa* for Production of Biomass and Secondary Metabolites

**DOI:** 10.3390/molecules27010014

**Published:** 2021-12-21

**Authors:** Nisar Ahmad, Palwasha Khan, Abdullah Khan, Maliha Usman, Mohammad Ali, Hina Fazal, Muhammad Nazir Uddin, Christophe Hano, Bilal Haider Abbasi

**Affiliations:** 1Centre for Biotechnology and Microbiology, University of Swat, Mingora 19200, Pakistan; palwashakhan2324@gmail.com (P.K.); khanbacha9999@gmail.com (A.K.); advasadullahkhan@gmail.com (M.U.); alimoh@uswat.edu.pk (M.A.); nazirkhattak@yahoo.com (M.N.U.); 2Pakistan Council of Scientific and Industrial Research (PCSIR) Laboratories Complex, Peshawar 25120, Pakistan; hina_fazalso@yahoo.com; 3Department of Agriculture, University of Swabi, Swabi 23460, Pakistan; drdurrishahwar@uoswabi.edu.pk; 4Laboratoire de Biologie des Ligneux et des Grandes Cultures (LBLGC), INRA USC1328, Université d’Orléans, CEDEX 2, 45067 Orleans, France; hano@univ-orleans.fr; 5Department of Biotechnology, Quaid-i-Azam University, Islamabad 45320, Pakistan

**Keywords:** *Stevia rebaudiana*, adventitious root culture, elicitation, *Cuscuta reflexa*, biomass

## Abstract

*Stevia rebaudiana* is an important medicinal plant that belongs to the *Asteraceae* family. The leaves of *Stevia rebaudiana* are a rich source of many health-promoting agents such as polyphenols, flavonoids, and steviol glycoside, which play a key role in controlling obesity and diabetes. New strategies such as the elicitation of culture media are needed to enhance the productivity of active components. Herein, the *Cuscuta reflexa* extracts were exploited as elicitors to enhance the productivity of active components. *Cuscuta reflexa* is one of the parasitic plants that has the ability to elongate very fast and cover the host plant. Consequently, it may be possible that the addition of *Cuscuta reflexa* extracts to adventitious root cultures (ADR) of *Stevia rebaudiana* may elongate the root more than control cultures to produce higher quantities of the desired secondary metabolites. Therefore, the main objective of the current study was to investigate the effect of *Cuscuta reflexa* extract as a biotic elicitor on the biomass accumulation and production of antioxidant secondary metabolite in submerged adventitious root cultures of *Stevia rebaudiana*. Ten different concentrations of *Cuscuta reflexa* were added to liquid media containing 0.5 mg/L naphthalene acetic acid (NAA). The growth kinetics of adventitious roots was investigated for a period of 49 days with an interval of 7 days. The maximum biomass accumulation (7.83 g/3 flasks) was observed on medium containing 10 mg/L extract of *Cuscuta reflexa* on day 49. As the concentration of extract increases in the culture media, the biomass gradually decreases after 49 days of inoculation. In this study, the higher total phenolics content (0.31 mg GAE/g-DW), total flavonoids content (0.22 mg QE/g-DW), and antioxidant activity (85.54%) were observed in 100 mg/L treated cultures. The higher concentration (100 mg/L) of *Cuscuta reflexa* extract considerably increased the total phenolics content (TPC), total phenolics production (TPP), total flavonoids content (TFC), total flavonoids production (TFP), total polyphenolics content (TPPC), and total polyphenolics production (TPPP). It was concluded that the extract of *Cuscuta reflexa* moderately improved biomass accumulation but enhanced the synthesis of phenolics, flavonoids, and antioxidant activities. Here, biomass’s independent production of secondary metabolites was observed with the addition of extract. The present study will be helpful to scale up adventitious roots culture into a bioreactor for the production of secondary metabolites rather than biomass accumulation in medicinally important *Stevia rebaudiana*.

## 1. Introduction

*Stevia rebaudiana* is one of the traditionally important plants worldwide. It belongs to the sunflower family (*Asteraceae*). It is commonly known as sweet leaf or sugar leaf due to its sweet taste [1]. The *Stevia rebaudiana* is famous for its sweet taste due to the chemical compounds present in the leaves of *Stevia* such as Steviol glycosides, which commonly includes steviosides and rebaudioside-A contents [2]. *Eupatorium rebaudianum Bertoni* was given the first name to *Stevia rebaudiana* [3]. The family *Asteraceae* has many species, but the *Stevia rebaudiana* contains the highest level of sweet compounds [4]. In early times, the Guarani tribes of Brazil used the *Stevia rebaudiana* leaves as medicines for heart burn in medicinal teas and also used it for other aliments [5]. *Stevia rebaudiana* is used for the treatment of hypotension, hypoglycemic condition, and its active compounds (Steviol glycosides) are naturally non-carcinogenic and one of the best sucrose substitutes [6,7]. It is also used to remove free radicals from the body and acts as a natural scavenger or antioxidant. Its leaves or active compounds can be used for curing heart diseases, and they are also helpful in controlling obesity and diabetes [8]. The steviol glycoside (stevioside) is 300 times sweeter than commercial sugar. It cannot enter into the bloodstream as it has no receptors and is therefore used as a sugar substitute for diabetic patients [1]. Steviol glycosides have no reported side effects such as genotoxicity and mutagenicity as compared to marketable sugar. The steviol glycosides also include other important active compounds such as dulcoside-A and rebaudioside-A-D contents [9].

The number of identified species shows that out of 150−200 *Stevia* species, only 20 species are cultivated throughout the world; these species are *S. nepetifolia*, *S. satureiaefilia*, *S. rhombifolia*, *S. rebaudiana*, *S. origanoides*, *S. triflora*, *S. lemmonii*, *S. plummerae*, *S. nepetifolia*, *S. serrata*, *S. tevia viscid*, *S. micrantha*, *S. myriadenia*, *S. oligophylla*, *S. eupatoria*, *S. selloi*, *S. ovata*, *S. leptophylla*, *S. ophryphylla*, and *S. commixta* [3]. *Stevia rebaudiana* is considered as having the highest sweetness level among these 20 species [10]. The approximate height of *Stevia rebaudiana* is 65 cm, having 3−4 cm ovate leaves, and they are stalk less [11]. *Stevia rebaudiana* have lilac, white-colored flowers [12]. Due to their natural ability of producing sweetness, they are also called the honey plant, sweet leaf, honey leaf, sweet herb, or sweet weed [13]. The optimum temperature for the ideal growth of *stevia* ranges from 15 to 30 °C with high rainfall. Due to its perennial nature, *Stevia* can also grow in a very cold areas having temperature below 0 °C [14]. The climate or habitat where *Stevia rebaudiana* could be easily found are forests, steppe climate, semi dry land, and also include grasslands [11]. *Stevia rebaudiana* culturing occurs in both semitropical and tropical countries [15]. *Stevia rebaudiana* is a good source of bulk stevioside, diterpene glycosides, rebaudioside A, rebaudioside B, rebaudioside C, etc. One of the qualities of *Stevia rebaudiana* is that it is a source of food, as it contains carbohydrates, vitamins, nonessential amino acids, polyphenols, and flavonoids, which play an important role in human health [1]. A number of sweet herbs/shrubs are present among the genus *Stevia*, but the sweetest member of this family is *Stevia rebaudiana*; apart from stevioside, a number of other sweetness link phytochemicals that have been isolated from their leaves are rebaudioside A, rebaudioside B, rebaudioside C, rebaudioside D, rebaudioside E, dulcoside A, and steviobioside [16].

The refined sugar has high sweetening content and high calories, from which may arise obesity and other persistent diseases, which include cardiovascular diseases, diabetes mellitus, and hypertension [17]. *Stevia rebaudiana* hold stevioside and rebaudioside A contents, which is confirmed as safe by the FDA, who defined that it is non-mutagenic, contains low calories, and is a non-toxic food ingredient in various countries such as Mexico, the UK, Korea, Indonesia, China, South America, Canada, Japan, and in U.S [18,19]. This important specie has been used by the Guarani tribes of Paraguay and Brazil as therapeutics “yerba mate” for heartburn from 15 centuries ago [5]. In addition to the sweetening properties of *Stevia rebaudiana*, more versatility is found in their natural properties, and it acts as anti-pathogenic [20], anti-cancerous [19], anti-hypersensitive [21], has radical scavenging property [21], and prevents tooth decay [19]. *Stevia rebaudiana* also plays an important role as a health mediator, having the properties of healing wounds, treating mouth sores and gums formation in lungs [22], and enhancing insulin production from the pancreas [22,23]. By the description of the FDA, in the human small intestine, there is no receptor for the absorption of stevioside; they are excreted out of the body without mixing with blood, so it can be used as an alternative of refined sugar for diabetic patients. By the declaration of the IDF, the total number of diabetic patients was 285 million in 2010 and will increase further to 439 million till 2030. So, by using the harmless *Stevia* extract as a sugar and carbohydrate source, it is possible to minimize the chances of diabetes in the increasing population [24]. *Stevia rebaudiana* plays a vital role in hypertension and obesity [25]. It has been confirmed that *Stevia* leaves control the diabetes in mice during a preclinical trial induced by streptozotocin (STZ), and they also help in curing liver and kidney problems [26]. Due to diabetes, the person may create an irregular level of lipid accumulation, irregular metabolism rate, insulin production from the beta cells of the pancreas, carbohydrate metabolism, and a difference in the levels of free radicals and antioxidants. Therefore, *Stevia* plays a key role in the regulation of these processes [27]. *Stevia rebaudiana* is an important component in food processing including fruit juices, baked products, tobacco derivatives, chewing gums, candies, pickles, coffee, and mostly beverages [28].

There are many methods used to cultivate *Stevia rebaudiana* such as the germination of seeds and sowing stem cuttings. The stem-cutting method propagates a smaller number of plants due the demand of primary stock in large quantities, which is very time consuming and season-dependent. The seed germination method also produces a minimum biomass of plantlets due to its small size, and sometimes, seeds lose the ability of germination (immature embryos) after collection [29]. Different methods are used to get the desired and valuable bioactive compounds and biomass in a short span of time. One of the biotechnological methods uses the plant cells, tissue, and organ culture, which conserve the endangered species as well as produce larger biomass in shorter duration [30,31]. Previously, many experiments have been done on the micropropagation of *Stevia rebaudiana* but little information is available regarding hairy and adventitious root cultures. Without field cultivation, adventitious root cultures give more drugs and bioactive compounds, which are useful for human health [32]. The adventitious root culture of *Stevia rebaudiana* and other medicinal plants will produce uniform quantities of secondary metabolites and are also easily growing as compared to conventional propagation. For the optimal biosynthesis of biomass and secondary metabolites, adventitious root culture is the best option because it is a useful method as compared to other in vitro methods. The reason is that the adventitious root culture takes a short time, grows easily, can be scaled up, and the culture conditions can be easily controlled such as chemical as well as physical conditions [33].

Elicitation is one of the important strategies that are used for the production of secondary metabolites in various valuable plants [34]. Biotic elicitors are those elicitors that are derived from living sources. Chitosan, glucans, chitin, pectin, cellulose, and glycoproteins are biotic elicitors of low molecular weight. In this study, *Cuscuta reflexa* was used as a biotic elicitor to enhance the biomass accumulation and production of secondary metabolites. *Cuscuta reflexa* is commonly known as devil’s hair and is also called a dodder plant. On the basis of Angiosperm phylogeny, it belongs to the morning glory family *Convolvulaceae* [34]. *Cuscuta reflexa* required a host plant for its survival. This plant takes nutrient sap from the host plant through vascular tissue for its growth and has no roots [35]. The dodder plant not only identifies its host plant, but it also has the ability to move toward its prey due to the volatile compounds of host plants which are released through the transpiration process [36]. *Cuscuta reflexa* seeds are used to treat vomiting and nausea. Antiviral, anticonvulsant activities, bradycardia, anti-spasmodic, anti-steroidogenic, and homodynamic activities are present in *Cuscuta reflexa* [37]. The seeds of *Cuscuta reflexa* are sweet in taste. The seeds of *Cuscuta reflexa* are used for the treatment of various disorders such as kidney and liver as well as used in the treatment of various diseases when used in combination with other medicinal plants. It controls the fluids that are lost from the body. For the treatment of jaundice, the juice of *Cuscuta reflexa* and *Saccharum officinarum* are used, and the paste of the plant is used for the treatment of headache [38]. The warm paste is used to treat rheumatism. Similarly, *Cuscuta reflexa* is also used for various disorders such as urination, muscle pain, and cough. It is also used to promote the growth of hair [39]. Various diseases can be treated by using *Cuscuta reflexa* such as body pains, constipation, itchy skin, flatulence, dry eyes, frequent urination, white discharge from the vagina, lower back pain, ringing in the ears, tired eyes, and blurred vision. It is also used as a blood purifier [40].

Therefore, the overall objective of the current study was to investigate the effect of *Cuscuta reflexa* as a biotic elicitor on growth behavior of adventitious root cultures of *Stevia rebaudiana*. Furthermore, the effect of *Cuscuta reflexa* as a biotic elicitor on polyphenolics content and antioxidant activity was also investigated. This study will provide a platform for future studies on using biotic elicitors for growth performance and the accumulation of secondary metabolites in highly valued medicinal plants.

## 2. Results and Discussion

### 2.1. Effect of Biotic Elicitor on Growth Kinetics of Adventitious Root Culture of Stevia

In this study, various concentrations of *Cuscuta reflexa* as a biological elicitor were applied to investigate the biomass accumulation in adventitious root culture of *Stevia rebaudiana*. The different concentrations of *Cuscuta reflexa* include 10, 20, 30, 40, 50, 60, 70, 80, 90, and 100 mg/3 flasks. Media having no extract of *Cuscuta reflexa* was used as control. The concentrations have shown a significant effect on biomass accumulation, as shown in Table 1 and Figure 1.

The data regarding biomass accumulation was calculated with 7-day intervals for a period of 49 days. In the current experiment, the addition of increasing concentrations of *Cuscuta reflexa* inhibited biomass accumulation in the adventitious root culture of *Stevia rebaudiana* (Figure 1). The highest biomass accumulation (8.16 mg/3 flasks) was observed in control culture on the 49th day of growth kinetics. When 10 mg/L *Cuscuta reflexa* extract was applied, it induced 2.97 g/3 flasks biomass after 7 days, but biomass gradually increased from 2.97 to 4.05 g/3 flasks (14 days), 5.13 g/3 flasks (21 days), 5.67 g/3 flasks (28 days), 6.2 g/3 flasks (35 days), 7.29 g/3 flasks (42 days), and finally reached 7.83 (49 days), after which the root started browning (Figure 1a). When 20 mg/L *Cuscuta reflexa* extract was applied, it induced 2.43 g/3 flasks biomass accumulation and gradually increased 2.43 to 3.5 g/3 flasks (14 days), 4.05 g/3 flasks (21 days), 5.13 g/3 flasks (28 days), 5.49 g/3 flasks (35 days), 6.12 g/3 flasks (42 days), and 7.29 g/3 flasks (49 days), which was comparatively lower than the T1 and control cultures (Table 1; Figure 1b). The addition of 30 mg/L *Cuscuta reflexa* extract induced 1.35 g/3 flasks biomass accumulation after 7 days, but biomass gradually increased from 1.35 to 2.16 g/3 flasks (14 days), 3.24 g/3 flasks (21 days), 4.32 g/3 flasks (28 days), 4.86 g/3 flasks (35 days), 5.13 g/3 flasks (42 days), and 5.94 g/3 flasks (49 days), as shown in Table 1 and Figure 1c. The application of 40 mg/L *Cuscuta reflexa* extract induced 1.08 g/3 flasks biomass accumulation after 7 days, but biomass gradually increased from 1.08 to 1.89 g/3 flasks (14 days), 2.403 g/3 flasks (21 days), 3.78 g/3 flasks (28 days), 4.59 g/3 flasks (35 days), 4.86 g/3 flasks (42 days), and 5.67 g/3 flasks (49 days), which was comparatively lower than T1, T2, and T3 and control cultures (Table 1; Figure 1d). When 50 mg/L *Cuscuta reflexa* extract was applied, it induced 0.99 g/3 flasks biomass biosynthesis after 7 days, but biomass gradually increased from 0.999 g/3 flasks to 1.512 g/3 flasks (14 days), 2.322 g/3 flasks (21 days), 2.7 g/3 flasks (28 days), 4.563 g/3 flasks (35 days), 4.59 g/3 flasks (42 days), and 5.184 g/3 flasks (49 days) and comparatively lowered the application of lower concentrations of *Cuscuta reflexa* extract and control (Table 1 and Figure 1e). When we applied 60 mg/L *Cuscuta reflexa* extract, it induced 0.81 g/3 flasks biomass gain after 7 days, but biomass gradually increased from 0.81 g/3 flasks to 1.35 g/3 flasks (14 days), 2.16 g/3 flasks (21 days), 2.43 g/3 flasks (28 days), 3.78 g/3 flasks (35 days), 4.59 g/3 flasks (42 days), and 5.86 g/3 flasks (49 days), which was comparatively lower than T1, T2, T3, T4, and T5 (10, 20, 30, 40 and 50 mg/L) and control cultures, as shown in Table 1 and Figure 1f.

The literature is limited to understanding the effect of biological elicitors such as *Cuscuta reflexa* on biomass regulation in liquid cultures of *Stevia*; however, other elicitors are widely reported that affect the biosynthesis of biomass and secondary metabolites. Polyethylene is one of the important elicitors that is commonly employed in plants as a stress inducer. In most cases, the PEG and paclobutrazol is used for growth retardation and to test the plant defense mechanism [41,42,43,44,45]. The combination of PEG and proline restricted the growth of *Stevia* cells and callus cultures. Similarly, the application of PEG negatively affected the shoot morphogenesis in *Stevia* [45]. Furthermore, PEG plays a key role in the transcription of six genes responsible for the synthesis of steviol glycoside [45]. PEG not only affects these genes but also limited the growth and development of *Stevia* plants in vitro. They subsequently noted that PEG negatively regulated UGT76G1, UGT74G1, and UGT85C2 of three UDP-dependent glycosyltransferases and other genes such as ent-kaurenoic acid hydroxylase, ent-kaurene synthase, and ent-kaurene oxidase. The quantities and concentrations of some high-valued secondary cell products are very rare in wild medicinal plants [36,45]. Plant cell, tissue, and organ cultures along with multiple elicitors provide an opportunity to enhance the productivity of secondary metabolites such as pharmaceuticals, cosmetics, nutraceuticals, and pigment as compared to wild plants [46]. For higher yield, the selection of proper species, optimization of the cultural conditions, physical or chemical elicitors, precursor feeding, and immobilization techniques are widely used for the optimal production of biomass and polyphenolics [47,48]. Plants in vitro cultures are commonly exploited because they do not require pesticides, herbicides, and weedicides; they are pathogen free; the light source and availability are uniform; there are constant culture conditions and no geographical variation, all of which lead to uniform productivity of the desired products in higher quantities such as in bioreactors [48]. The in vitro cultures are trustworthy for the production of biomass and secondary metabolites due to the homogenous growth of cells in contrast to wild plants, which are exposed to multiple and variable environmental conditions that may cause fluctuation in the synthesis of biomass and important metabolites [49]. Furthermore, the selection of a proper explant also plays a key role in culture development and the synthesis of metabolites because the polyphenolics are not restricted to specific parts in wild species but rather multiple organs of wild plants [1,2]. In previous studies, the exposure of *Stevia* in vitro cultures to 0, 20, 40, 60, and 80 mM concentrations of salt (NaCl) negatively regulated the growth of *Stevia* plants [50]. The multiple concentrations of salt also affected the *Stevia* plant morphology and also observed that the *Stevia* plant is less tolerant to salt; however, the salt enhanced the biosynthesis of Steviol glycosides compared with the control cultures [51].

Ahmad et al. [52] applied differential pH levels that range from 5.0 to 6.0 and observed varied results during the development of adventitious root cultures of *Stevia*. They further observed that the adjustment of culture media with a pH level of 6.0 boosted the accumulation of fresh and dry biomass (112.86 g/L^−1^ and 8.29 g/L^−1^) after 30 days of roots inoculation. During lag, log, and decline phases, the maximum adventitious root biomass of 112.5 g L^−1^ is recorded on the 27th day of log phases during growth kinetics. Gold (Au) and copper (Cu) nanoparticles (NPs) play a key role in the biosynthesis of biomass and low molecular weight secondary metabolites in *S. rebaudiana* [53]. The addition of selective ratios of copper and gold nanoparticles significantly enhance the accumulation of biomass and secondary cell products in adventitious root cultures of *Stevia*. The exposure of inoculum roots in suspension media augmented with AuCu (1:2) nanoparticles displayed the highest accumulation of biomass (1.447 g/flask) on the 27th day of log phases of growth kinetics [53]. Other ratios of Cu and Au NPs also played an important role in the synthesis of biomass and secondary metabolites.

When 70 mg/L *Cuscuta reflexa* extract was applied on ADR cultures of *Stevia*, it induced 0.54 g/3 flasks biomass after 7 days, but biomass gradually increases from 0.54 g/3 flasks to 1.08 g/3 flasks (14 days), 2.052 g/3 flasks (21 days), 2.322 g/3 flasks (28 days), 3.267 g/3 flasks (35 days), 4.59 g/3 flasks (42 days), and finally reached 4.806 g/3 flasks (49 days), as shown in Table 1 and Figure 1g. When 80 mg/L *Cuscuta reflexa* extract was applied, it induced 0.27 g/3 flasks biomass accumulation after 7 days, but biomass gradually increases from 0.27 g/3 flasks to 1.08 g/3 flasks (14 days), 1.863 g/3 flasks (21 days), 2.16 g/3 flasks (28 days), 2.97 g/3 flasks (35 days), 3.969 g/3 flasks (42 days), and finally reached 4.509 g/3 flasks (49 days), as shown in Table 1 and Figure 1h. The addition of 90 mg/L *Cuscuta reflexa* extract to culture media induced 0.27 g/3 flasks biomass gain after 7 days, but biomass gradually increases from 0.27 g/3 flasks to 0.54 g/3 flasks (14 days), 1.728 g/3 flasks (21 days), 2.106 g/3 flasks (28 days), 2.511 g/3 flasks (35 days), 3.51 g/3 flasks (42 days), and the final biomass was 4.05 g/3 flasks after 49 days period (Table 1 and Figure 1i). When we applied 100 mg/L *Cuscuta reflexa* extract, it induced 0.27 g/3 flasks biomass after 7 days, but biomass gradually increases from 0.27 g/3 flasks to 0.54 g/3 flasks (14 days), 1.35 g/3 flasks (21 days), 1.863 g/3 flasks (28 days), 2.268 g/3 flasks (35 days), 3.321 g/3 flasks (42 days), and the final biomass gain was 3.726 g/3 flasks after a 49-day period; afterwards, the cultures started browning, as shown in Figure 1j (Table 1). It shows that the increasing concentration of *Cuscuta reflexa* inhibited biomass accumulation in adventitious root culture of *Stevia rebaudiana.* After the completion of the experiment, the overall roots fresh and dry weights were determined as shown in Table 2.

Among multiple elicitors, gibberellic acid is considered as a biotic/chemical elicitor that plays a key role in biomass gain and active compounds production in *Stevia*. Gibberellic acid (GA) is one the potent hormones that showed suitable results on plant growth, biomass gain, and morphogenesis in *Stevia* [54,55]. GA also played an important role in cultures development of highly medicinal plants. The addition of various concentrations of GA improved the biosynthesis of secondary metabolites and biomass of hairy root of Datura and *Echinacea purpurea* [30,56]. The application of GA showed different results in different plant species [56,57]. Furthermore, Ahmad et al. [58] performed a study on the effects of various spectral lights for the formation of secondary metabolites and accretion of biomass in *Stevia* callus culture. Furthermore, Ahmad et al. [58] observed that yellow light improved callus formation than other colored lights, while the maximum callogenesis was observed under white light. In another study, the production of secondary metabolites and accumulation of biomass in adventitious root cultures of *S.*
*rebaudiana* have been investigated under different spectral lights. Phenolics (102.32 µg/g DW), flavonoids (22.07 µg/g DW), and antioxidant potential (11.63 µg/g DW) were found to be higher in root cultures grown under blue light [59]. During growth kinetics, the highest fresh root biomass was attained in cultures placed under violet light on day 27 in the log phase. However, green, yellow, blue, control, and red lights also produced optimal biomass in log phases. The current results conclude that violet light is more effective for biomass accumulation than other colored lights as well as control.

### 2.2. The Effect of Cuscuta Reflexa Extract on Polyphenolics in Adventitious Roots Cultures

In current study, phenolic accumulation was observed in response to 10 different concentrations (10, 20, 30, 40, 50, 60, 70, 80, 90, and 100 mg/L) of *Cuscuta reflexa* extracts as an abiotic elicitor. Herein, 10 mg/L concentration of *Cuscuta reflexa* extract exhibited the accumulation of 0.06 GAE-mg/g-DW total phenolic content (TPC) on day 49. As the concentration of extracts increased, the biosynthesis of phenolics contents also increased. After the addition of 100 mg/L extract to culture media, the phenolic accumulation was observed as 0.31 GAE-mg/g-DW on day 49. In this study, the highest value was 0.31 GAE-mg/g-DW at a concentration of 100 mg/L as compared to the control group that produced 0.013 GAE-mg/g-DW without any stress, as shown in Table 3. In the current study, *Cuscuta reflexa* showed a significant effect on TFC. TFC increases with increasing the concentration of *Cuscuta reflexa* in the culture media. When 10 mg/L extract was added to culture media, the TFC accumulation was 0.01 mg QE/g DW; however, the biosynthesis of flavonoids increased, similar to TPC biosynthesis. The highest TFC (0.22 mg/g-QE/g DW) was accumulated with the addition of 100 mg/L extract of *Cuscuta reflexa* as compared to the control group that is 0.11 mg/g-QE/g DW, as shown in Table 3. The literature regarding the effect of abiotic elicitors on polyphenolics content in *Stevia* is very limited; however, the effects of other elicitors on plant development and the production of secondary metabolites are widely reported. Among various elicitors, polyamines such as putrescene, spermidine, and spermine play a key role in plant development. The elicitation of culture media with polyamines significantly enhances cellular differentiation, biomass gain, and the synthesis of active compounds in *Stevia*. The cationic nature of (PAs) intermingle with nucleic acid, protein, membrane, and other molecules and subsequently increase or activate other enzymes/chemicals that directly or indirectly enhance the process of morphogenesis [60,61]. The multiple concentrations of various polyamines also play an important role in the biosynthesis of biomass and bioactive compounds. It is considered one of the important hormonal messenger and growth regulators that fluctuate the biochemical pathways of different cells and tissues and also are involved in the biosynthesis of important chemicals such as rosmarinic acid, esculin, betalaine, esculetin, and coumarin in medicinally important plant species [62,63].

In the current study, 0.0504 g/3 flasks of total phenolics production (TPP) were observed in roots treated with 10 mg/L extract of *Cuscuta reflexa*, 0.0740 g/3 flasks were observed in roots treated with 20 mg/L extract of *Cuscuta reflexa*; however, as the concentrations of the extract increases, the production also increased, and 0.1457 g/L was observed by applying 100 mg/L extract of *Cuscuta reflexa*, as shown in Table 3. The application of 10 mg/L extract of *Cuscuta reflexa* resulted in 0.0084 g/L of total flavonoids production (TFP). As the concentration of the extract increased, the TFP also increased. At last, 0.1025 g/L TFP was observed by applying 100 mg/L extract of *Cuscuta reflexa*, as shown in Table 3. Furthermore, the application of extracts also enhanced the synthesis of total polyphenolics content (TPPC). Herein, 10 mg/L extract of *Cuscuta reflexa* induced 0.07 g/L TPPC, 20 mg/L extract of *Cuscuta reflexa* gave 0.14 g/L, 30 mg/L gave 0.24 g/L, 0.109 g/L was observed by applying 40 mg/L extract of *Cuscuta reflexa*, 0.316 g/L was observed by applying 50 mg/L extract of *Cuscuta reflexa*, 0.37 g/L was observed by applying 60 mg/L extract of *Cuscuta reflexa*; then, by applying 70 mg/L extract of *Cuscuta reflexa*, it gave 0.441 g/L TPPC. Then, 0.482 g/L was noted by using 80 mg/L extract of *Cuscuta reflexa*, 0.499 g/L by 90 mg/L, while 0.528 g/L was observed by applying 100 mg/L extract of *Cuscuta reflexa*, as shown in Table 3. In the current study, 0.0588 g/L of total polyphenolics production (TPPP) was observed by applying 10 mg/L extract of *Cuscuta reflexa*. Moreover, 20 mg/L extract of *Cuscuta reflexa* exhibited 0.1036 g/L TPPP, 30 mg/L extract of *Cuscuta reflexa* gave 0.1968 g/L TPPP, 0.06976 g/L of TPPP was determined by applying 40 mg/L extract of *Cuscuta reflexa*, 0.22752 g/L was determined by 50 mg/L extract of *Cuscuta reflexa*, 0.2294 g/L was observed by the application of 60 mg/L, 0.2646 g/L was observed by applying 70 mg/L, 0.32776 g/L was observed by applying 80 mg/L, then 0.32934 g/L was observed by applying 90 mg/L, while 0.22176 g/L was determined by applying 100 mg/L extract of *Cuscuta reflexa*, as shown in Table 3.

The production of such important compounds can be increased by the elicitation strategies. The exposure of any in vitro cultures to biotic or abiotic elicitors/stressors positively regulates the biosynthesis of high-valued secondary cell products by altering the metabolic pathways [48]. Multiple elicitors are commonly exploited to enhance the metabolites of interest in various in vitro cultures of pharmaceutically important plants [64,65]. Some of the commonly used elicitors that are used for maximum secondary metabolites production include chitosan, pectin, salicylic acid, alginate, casein hydrolysate, methyl jasmonate, and extract of yeast [66]. Different elicitors produce different quantities of secondary metabolites depending on plant species [36,45]. However, the main objective of the addition of elicitor to any culture media is to obtain a uniform and higher yield of active compounds [49,67]. Elicitors in combination with auxin/cytokine play an important role in the productivity of final pharmaceutical products [48,68]. Moreover, these elicitors directly or indirectly influence the metabolic pathways of secondary metabolites and either decrease or increase the production depending upon the culture conditions and optimization process and the selection of proper biological or non-biological elicitors [58,69]. The productivity of secondary metabolites biosynthesis depends upon the addition of additives to culture media, aeration, type of culture medium, agitation, source and type of carbohydrates, level of phosphate and nitrate, type of plant hormones, oxygen supply, nutrients feeding, carbon source other than carbohydrate, and quantity, intensities, and exposure time of light [70]. Such an alteration of culture media with different elicitors produces ROS, which directly interact with biological molecule and finally fluctuate the synthesis of secondary metabolites and plant development [65]. In response to ROS, plants activate the defense mechanism and either release enzymatic or non-enzymatic components (polyphenolics, ascorbate, tocopherol, glutathione, superoxide dismutase, and peroxidases) that neutralize the damaging effect of ROS but increase the release of secondary cell products that are the actual metabolites needed for various medicinal uses [71]. Similar reports have been published on the development of adventitious roots in medicinally important species of *Plumbago zeylanica*, *Eurycoma longifolia*, and *Centella asiatica* for the production of highly valued secondary metabolites [72,73,74,75,76].

### 2.3. DPPH Radical Scavenging Activity (DRSA)

In the current study, antioxidant activity was determined as percentage of DPPH (2,2-Diphenyl-1-Picryl Hydrazyl) radical scavenging activity (DRSA) in different treated samples. When 10 mg/L extract was added, the activity was 54.2%; when 20 mg/L extract was added to culture media, the activity was 57.7%, 30 mg/L extract induced 61.6% activity, 40 mg/L extract addition exhibited 67.9% activity, 50 mg/L extract exhibited 72.1% activity, 60 mg/L extract addition showed antioxidant activity of 77.03%, 70 mg/L extract induced 79.88% antioxidant activity, 80 mg/L extract exhibited 81.33% activity, 90 mg/L extract exhibited 84.76% activity, while the addition of 100 mg/L extract exhibited the activity of 85.54%, as shown in Table 4. It was illustrated that when the concentration of *Cuscuta reflexa* increases, the activity also increased.

The intact leaves or its medicinal derivatives have proven antioxidant, antimicrobial, antihyperglycemic, anticancerous, and antihypersentitive properties [19,21,22]. It has been reported that the foliar application of GA significantly enhances the antioxidant potential in *Stevia*. Ahmad et al. [59] observed that yellow light improved callus formation compared with other colored lights, while the maximum callogenesis was observed under white light. They observed that culture under blue light enhances phenolics, flavonoids, and antioxidant potential. It means that blue light is one of the best candidates for the biosynthesis of secondary metabolites in calli cultures of *Stevia rebaudiana*. However, up to some extent, red and green lights also increase DRSA (80%) [59]. In another study, the phenolics (102.32 µg/g DW), flavonoids (22.07 µg/g DW), and antioxidant potential (11.63 µg/g DW) were found to be higher in root cultures grown under blue light [59]. Ahmad et al. [53] observed that a 5.1 pH level enhanced the biosynthesis of steviosides and rebaudioside (79.48 and 13.10 mg/g-DW), while a pH level of 5.8 improved dulcoside content (2.57 mg/g-DW). They further reported that a pH level of 5.8 enhanced the production of phenolics, flavonoids, and antioxidant potential (70.06 and 50.19 mg/g-DW, 92.67%) in adventitious root cultures of *Stevia rebaudiana.*

## 3. Materials and Methods

### 3.1. Seeds Collection and Germination

Seeds were collected from The University of Agriculture, Peshawar. Surface decontamination was done according to the protocol of Ahmad et al. [37]. After applying 70% ethanol (Merk, Kenilworth, NJ, USA) and mercuric chloride (HgCl_2_; 0.2%; Sigma Aldrich, Saint Quentin Fallavier, France), the surface-sterilized seeds were washed many times using autoclave-distilled water to remove the remaining toxic chemicals (if any). For the germination of seeds, Murashige and Skoog medium (MS, Phyto-Tech, Lenexa, KS, USA [77]) was prepared having no plant growth regulators. For the preparation of medium, agar was used (7−8 g/3 flasks) as a solidifying agent, media having an optimum pH of 5.8, 30 g/L sucrose was used and then kept in an autoclave for 20 min at 121 °C under 15 psi pressure. The seeds were inoculated onto sterilized MS media, and the cultured flasks were transferred to growth chambers having a controlled environment (40-mol m^−2^, 25 ± 2, and 16/8 h photoperiod). The cultures were placed in a growth room for a period of 30 days. For the development of adventitious root cultures, explants were collected after 30 days of seed inoculation and germination.

### 3.2. Explant Collection for Development of Stock Adventitious Root Cultures

Fresh roots (~1–2 cm) were collected as explant from in vitro grown plantlets for the development of adventitious stock cultures. In 100 mL flasks, the roots were grown, which were obtained from in vitro plantlets. Media were comprised of 40 mL MS-basal media having different concentrations of NAA (0.5–2.0 mg/L) according to the established protocols of Ahmad et al. [1,53]. In an orbital shaker, these flasks were kept (120 rpm; Gallankamp, London, England) at 25 °C for 7 weeks until the formation of adventitious roots cultures.

### 3.3. Establishment of Adventitious Root Cultures

Different concentrations of NAA (Sigma Aldrich, Saint Quentin Fallavier, France) were added to MS media for the development of stock cultures and to identify the best concentration of NAA for maximum biomass development. However, we also kept in mind the previous established protocols of Ahmad et al. [1,53] for *Stevia* ADR development, in which the 0.5 mg/L exhibited maximum biomass production. Adventitious roots were used as inoculum (≈1−2 cm), which is then transferred to MS media. After 7 weeks, adventitious roots appeared from roots segments. MS media was prepared having five different treatments as T1 (0.5 mg/L), T2 (1.0 mg/L), T3 (1.5 mg/L), T4 (2.0 mg/L), and control. Here, the best response was indicated by the application of 0.5 mg/L NAA; therefore, NAA (0.5 mg/L) was present in all treatments (Ahmad et al., 2018, 2021). The constant concentration of NAA (0.5 mg/L) was combined with *Cuscuta reflexa* extract having different concentrations as T1: 10 mg/3 flasks, T2: 20 mg/3 flasks, T3: 30 mg/3 flasks, T4: 40 mg/3 flasks, T5: 50 mg/3 flasks, T6: 60 mg/3 flasks, T7: 70 mg/3 flasks, T8: 80 mg/3 flasks, T9: 90 mg/3 flasks, and T10: 100 mg/3 flasks and control media. These treatments were incubated for a period of 7 weeks at 120 rpm on an incubator orbital shaker (Gallenkamp, London, England) at 25 °C. Growth kinetics data were recorded with an interval of 7 days for a 49-day period. In response to different concentrations of *Cuscuta reflexa* extracts (T1–T10: 10 mg/3 flasks to 100 mg/3 flasks) with constant NAA (0.5 mg/L) concentration, the growth curve was established for rapidly growing root cultures.

### 3.4. Adventitious Root Biomass Determination

For the investigation of fresh biomass, the adventitious roots of each treatment (T1–T10) were collected from liquid media. For the elimination of media particles, the roots were washed with sterile distilled water, and fresh weight was determined. Furthermore, by pressing gently with sterilized filter paper (Waltman) to remove the excess of water, subsequently, the roots were oven dried (Thermo Scientific, Dreieich, Hessen, Germany) at 45−55 °C and then weighed for dry weight determination. Finally, the fresh and dry biomass of adventitious roots (ADR) were presented in grams per 3 flasks.

### 3.5. Analytical Methods

#### 3.5.1. Extract Preparation

The extract preparation was carried out according to the methods of Fazal et al. [34]. Here, exactly 0.4 g from each treatment (T1–T10) was ground with the help of a grinder. The powdered samples of each treatment were transferred to air-tight vials. Accurately, 1 mg of powdered material of each treatment was added to a test tube and then blended with 10 mL of HPLC grade ethanol (99%; Merk, Kenilworth, NJ, USA). For 1 week, the solution was vortexed daily to secrete maximum secondary metabolites to the ethanol solvent. Then, the solution was centrifuged for 15 min at 14,000× *g* rpm. To find the different activities, the supernatant layer was taken and was used for further activities.

#### 3.5.2. Total Phenolics Content (TPC)

The method of Ahmad et al. [37] was used for the investigation of total phenolics content in each treated sample. Briefly, 2.5 mL of 2 N Folin–Ciocalteu reagent (FCR; Sigma Aldrich, Saint Quentin Fallavier, France) was combined with each treated extract (0.1 mL) followed by incubation for four min and then mixed with 2.55 mL of distilled water. The solution was finally mixed with 20% Na_2_CO_3_ (Merk, Kenilworth, NJ, USA) before the incubation for a period of 1 h in dark (to avoid other reactions such as unwanted oxidation). For 14 min, the mixture was centrifuged at 10,000× *g* rpm. The mixture was filtered through a 45-µm membrane (Merk) in a UV-visible spectrophotometer cuvette, and the absorbance of the resulted mixture was measured at 760 nm. For plotting a standard calibration curve, we used gallic acid (Sigma 1.0−10 ug/mL; *R*^2^ = 0.9889), as shown in Figure 2. The results were expressed as gallic acid equivalent (GAE) mg/g of dry root biomass. The following Equation (1) was used to determine TPC.
% Total phenolics content = 100 × (AS − AB) / (CF × DF)(1)

Here, “AS” and “AB” represent the absorbance of each treated sample (extract) and the absorbance of 99% ethanol as blank; while the “CF” represents the conversion factor and “DF” is the dilution factor for the solution at a concentration of 0.1 mL. Furthermore, the phenolics production was determined by the multiplication of TPC with the dry weight of each treated sample (T1–T10).

#### 3.5.3. Total Flavonoids Content (TFC)

For the determination of TFC in ADR, the protocol of Ahmad et al. [37] was followed. Ethanol extract (0.25 mL) of each treatment was mixed with AlCl_3_ [(0.075 mL) 5% (*w*/*v*); Merk; Kenilworth, NJ, USA], and 1.25 mL of sterile distilled water and finally 0.5 mL of NaOH (1 M; Merk; Kenilworth, NJ, USA) was mixed with the solution and incubated for 1 h in dark to avoid oxidation of the reaction mixture. Subsequently, for 14 min, the solution was centrifuged at 10,000× *g* rpm to obtain the desired supernatant containing the metabolites of interest. A standard calibration curve was established for different concentrations of quercetin (Sigma; 1.0–10 mg/mL; *R*^2^ = 0.9866), and a UV-visible spectrophotometer was used to check the absorbance of each treated sample as well as for different concentrations of quercetin (QE) at 510 nm wavelength. The results for each treated sample were expressed as mg/g-QE-DW, and the TFC was finally obtained by using the following Equation (2):% Total Flavonoids content = 100 × (AS − AB) / (CF × DF)(2)

Here, “AS” and “AB” represent the absorbance of each treated sample (extract) and the absorbance of 99% ethanol as blank, while the “CF” represents the conversion factor and “DF” is the dilution factor for the solution at a concentration of 0.1 mL. Furthermore, the flavonoids production was determined by the multiplication of TFC with the dry weight of each treated samples (T1–T10).

#### 3.5.4. DPPH Radical Scavenging Activity (DRSA)

For the determination of DRSA, 2,2-Diphenyl-1-Picryl Hydrazyl (DPPH; Sigma Aldrich, Saint Quentin Fallavier, France) powder (0.25 mg) was dissolved in 99% HPLC grade ethanol (20 mL) as solution A. Each treated sample was indicated as solution B. Here, solution “B” 2.0 mL of DPPH free radical solution (0.25 mg/20 mL × 4) was mixed with the ethanol extract of each sample (solution A; 1 mL) according to the method of Ahmad et al. [78]. Then, it was placed in a dark room for a period of half an hour to avoid oxidation. To test the absorbance of the resulted mixture, a wavelength of 517 nm was used. Then, the following Equation (3) was applied for DPPH assay.
DRSA (%) = 100 × (1 − AP/AD)(3)

In this above equation, AD represents the absorbance of DPPH solution having no tissue extract, and AP represents the absorbance of extract at 517 nm.

### 3.6. Statistical Analysis

An independent experiment was used for each treatment and revised twice. For the determination of mean values, one-way analysis of variance (ANOVA) was applied. An Excel sheet exploited the data arrangement and percentage determination. Mean values with standard deviation, least significant differences, and probability were determined using Statistix software v.8.2 (USA).

## 4. Conclusions

In the current study, it was concluded that when different concentrations of *Cuscuta reflexa* were added to liquid media, the biomass accumulation decreases while the secondary metabolites such as phenolics, flavonoids, and antioxidant activities were increased. Among various extracts of *Cuscuta reflexa* concentrations, the 100 mg/L was found to be effective in fresh biomass accumulation after 49 days of inoculation. Similarly, 100 mg/L extract of *Cuscuta reflexa* was effective for TFC, TPC, as well as DPPH activity.

In the current study, the effect of different concentrations (10, 20, 30, 40, 50, 60, 70, 80, 90, and 100 mg/L) of *Cuscuta reflexa* on the accumulation of phenolics was determined. It was observed that 10, 20, 30, 40, 50, 60, 70, 80, 90, and 100 mg/L extract of *Cuscuta reflexa* resulted in 0.06, 0.1, 0.17, 0.189, 0.199, 0.21, 0.25, 0.279, 0.292, and 0.31 mg GAE/g DW of TPC production. However, the highest value was obtained with the exposure to 100 mg/L extract of *Cuscuta reflexa* (control 0.013 mg GAE/g DW on day 49). This study showed that the extract of *Cuscuta reflexa* increases and affects the production of secondary metabolites. The effect of *Cuscuta reflexa* extract on TFC was directly proportional to the used concentrations: i.e., the lower the concentration, the lower the TFC, while the higher the concentration, the higher the TFC biosynthesis. Cultures treated with 10, 20, 30, 40, 50, 60, 70, 80, 90, and 100 mg/L extract of *Cuscuta reflexa* resulted in 0.01, 0.04, 0.07, 0.92, 0.117, 0.16, 0.191, 0.203, 0.207, and 0.218 mg QE/g DW on day 49. The highest TFC accumulation was observed by the addition of 100 mg/L extract of *Cuscuta reflexa*. In the current study, the data showed that DRSA activity increases with the increasing concentration of extract of *Cuscuta reflexa*, and at 100 mg/L extract of *Cuscuta reflexa*, it was determined as 85.54 g/L, while at 10 mg/L extract of *Cuscuta reflexa*, it was observed as 54.2 g/L. In another study, GA_3_ was used to enhance the production of tanshinones and caffeic acid derivatives (CADs) in *S. miltiorrhiza* hairy roots and *Echinacea purpurea* hairy roots [30,79,80,81,82]. In other studies, the artemisinin accumulation in root cultures of *A. annua* was enhanced by GA_3_ hormones [81,82]. In hairy root culture of *S. miltiorrhiza*, it was found that the production of phenolics increases by using GA_3_ [82]. In another study, it was found that GA_3_ affects the production of secondary metabolites by either increasing or decreasing the concentration of compounds produced [83,84]. In the current study, the maximum antioxidant activity in adventitious roots cultures was 85.54% exposed to 100 mg/L extract of *Cuscuta reflexa,* followed by 90 mg/L, 80 mg/L, 70 mg/L, 60 mg/L, and 50 mg/L extract of *Cuscuta reflexa*, which enhances the antioxidant activity to 84.76%, 81.33%, 79.88%, 77.03%, and 72.1%, which is comparatively lower than 100 mg/L extract of *Cuscuta reflexa* but higher than the control. The effect of non-biological elicitors has been reported on biomass accumulation in an adventitious root culture of *Stevia rebaudiana*, but the literature is limited regarding the effect of biological elicitors on the growth of adventitious root cultures of *Stevia*. This is the first study on the effect of biological elicitor on biomass accumulation of adventitious root culture of *Stevia rebaudiana*. Therefore, the current study observed biomass independent production of secondary metabolites and moderate improvement in biomass accumulation. Furthermore, the current study suggested that fresh extract should be more valuable than dry extract application. The *Cuscuta reflexa* extract did not show positive regulation in biomass accumulation but enhanced the biosynthesis of secondary metabolites.

## Figures and Tables

**Figure 1 molecules-27-00014-f001:**
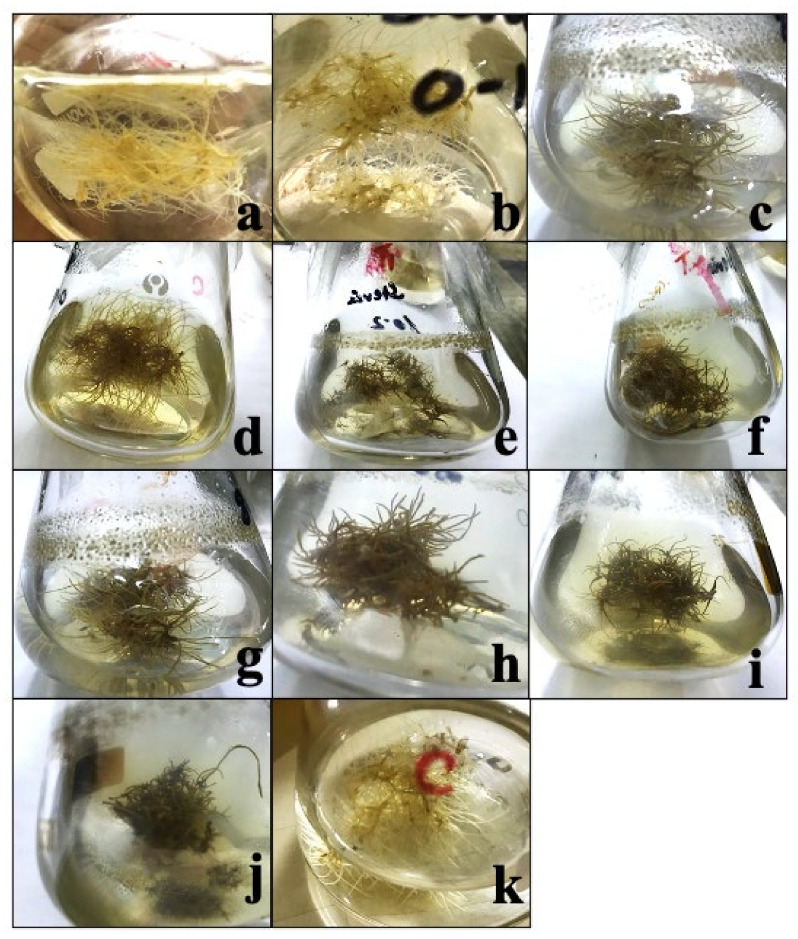
Pictorial presentation of the effect of 10 different extracts: (**a**) 10 mg/3 flasks, (**b**) 20 mg/3 flasks, (**c**) 30 mg/3 flasks, (**d**) 40 mg/3 flasks, (**e**) 50 mg/3 flasks, (**f**) 60 mg/3 flasks, (**g**) 70 mg/3 flasks, (**h**) 80 mg/3 flasks, (**i**) 90 mg/3 flasks, (**j**) 100 mg/3 flasks, and (**k**) control of *Cuscuta reflexa* on the adventitious root culture development of *Stevia rebaudiana*.

**Figure 2 molecules-27-00014-f002:**
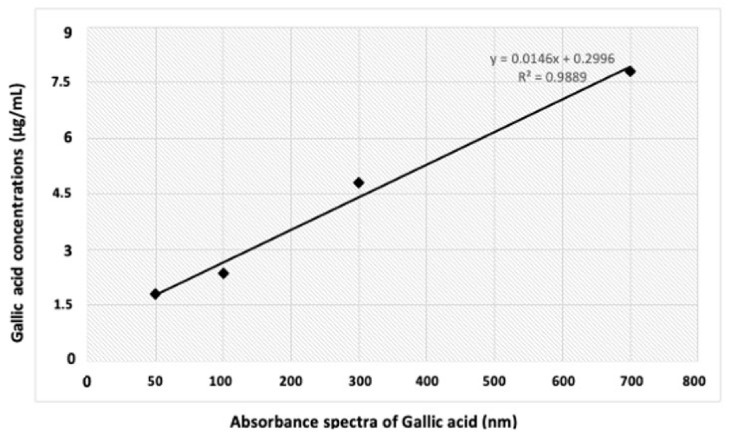
Standard calibration curve for gallic acid, where “y” is the absorbance spectra of each concentration, “x” represents the concentration of gallic acid (Sigma; 1.0−10 µg/mL), and “*R*^2^” represents the correlation coefficient. A similar plot was also calibrated for the different concentrations of quercetin to compare the treated samples for the investigation of flavonoids contents where the correlation coefficient (*R*^2^) values fall within in the range of 0.9866.

**Table 1 molecules-27-00014-t001:** Effect of different *Cuscuta reflexa* (*Cr*) extracts on the growth kinetics of adventitious roots biomass (ADR) of *Stevia rebaudiana*. Data were collected with 7-day intervals for a period of 49 days. Mean values with standard deviation (±SD) of the means did not exceed 5%. Mean values for each treatment were obtained from three independent experiments. Mean values with common alphabets are significantly different at *p* < 0.05.

Culture Period	7 Days	14 Days	21 Days	28 Days	35 Days	42 Days	49 Days
	ADR Biomass (g/3 Flasks)	ADR Biomass (g/3 Flasks)	ADR Biomass (g/3 Flasks)	ADR Biomass (g/3 Flasks)	ADR Biomass (g/3 Flasks)	ADR Biomass (g/3 Flasks)	ADR Biomass (g/3 Flasks)
*Cr* (10 mg/3 flasks)	2.97 ± 0.01 c	4.05 ± 0.05 b	5.13 ± 0.03 ab	5.67 ± 0.01 ab	6.2 ± 0.1 a	7.29 ± 0.09 a	7.83 ± 0.08 a
*Cr* (20 mg/3 flasks)	2.43 ± 0.01 b	3.5 ± 0.03 b	4.05 ± 0.02 ab	5.13 ± 0.03 ab	5.49 ± 0.08 ab	6.12 ± 0.05 a	7.29 ± 0.08 a
*Cr* (30 mg/3 flasks)	1.35 ± 0.02 c	2.16 ± 0.04 b	3.24 ± 0.03 b	4.32 ± 0.02 ab	4.86 ± 0.03 ab	5.13 ± 0.02 a	5.94 ± 0.02 a
*Cr* (40 mg/3 flasks)	1.08 ± 0.01 c	1.89 ± 0.01 c	2.41 ± 0.01 b	3.78 ± 0.03 b	4.59 ± 0.02 ab	4.86 ± 0.05 a	5.67 ± 0.04 a
*Cr* (50 mg/3 flasks)	0.99 ± 0.01 c	1.52 ± 0.01 b	2.32 ± 0.01 b	2.7 ± 0.02 b	4.56 ± 0.02 a	4.6 ± 0.04 a	5.2 ± 0.03 a
*Cr* (60 mg/3 flasks)	0.81 ± 0.01 c	1.35 ± 0.01 c	2.16 ± 0.01 bc	2.43 ± 0.02 bc	3.78 ± 0.03 b	4.59 ± 0.06 a	5.86 ± 0.04 a
*Cr* (70 mg/3 flasks)	0.54 ± 0.01 c	1.08 ± 0.01 b	2.05 ± 0.01 b	2.32 ± 0.01 b	3.27 ± 0.0 ab	4.6 ± 0.03 a	4.8 ± 0.02 a
*Cr* (80 mg/3 flasks)	0.27 ± 0 c	1.08 ± 0.01 bc	1.86 ± 0 bc	2.2 ± 0.01 b	2.9 ± 0.02 b	3.97 ± 0.03 b	4.5 ± 0.03 a
*Cr* (90 mg/3 flasks)	0.27 ± 0 c	0.54 ± 0 c	1.73 ± 0 bc	2.1 ± 0.01 b	2.5 ± 0.02 b	3.5 ± 0.01 ab	4.1 ± 0.02 a
*Cr* (100 mg/3 flasks)	0.27 ± 0 c	0.54 ± 0 bc	1.4 ± 0 b	1.86 ± 0.01 b	2.3 ± 0.01 ab	3.3 ± 0.03 a	3.74 ± 0.03 a
Control	0.34 ± 0 e	1.17 ± 0 d	3.27 ± 0 cd	5.33 ± 0.01 c	6.94 ± 0.01 b	7.77 ± 0.02 ab	8.16 ± 0.01 a

**Table 2 molecules-27-00014-t002:** Effect of different *Cuscuta reflexa* extracts on the fresh and dry weights of adventitious root cultures of *Stevia rebaudiana* after the completion of the experiments. Mean values with standard deviation (±SD) of the means did not exceed 5%. Mean values for each treatment were obtained from three independent experiments. Mean values with common alphabets are significantly different at *p* < 0.05.

Plant Extracts	Fresh Weight (g/3 Flask)	Dry Weight (g/3 Flask)
*C. reflexa* (10 mg/3 flasks)	9.14 ± 1.2 a	0.84 ± 0.1 a
*C. reflexa* (20 mg/3 flasks)	8.9 ± 0.7 ab	0.74 ± 0.02 ab
*C. reflexa* (30 mg/3 flasks)	8.84 ± 0.3 ab	0.82 ± 0.02 a
*C. reflexa* (40 mg/3 flasks)	8.66 ± 0.1 ab	0.64 ± 0.03 ab
*C. reflexa* (50 mg/3 flasks)	8.14 ± 0.3 ab	0.72 ± 0.01 ab
*C. reflexa* (60 mg/3 flasks)	8.04 ± 0.7 ab	0.62 ± 0.04 ab
*C. reflexa* (70 mg/3 flasks)	7.4 ± 0.2 b	0.6 ± 0.03 ab
*C. reflexa* (80 mg/3 flasks)	7.54 ± 0.2 b	0.68 ± 0.02 ab
*C. reflexa* (90 mg/3 flasks)	7.26 ± 0.3 b	0.66 ± 0.01 ab
*C. reflexa* (100 mg/3 flasks)	5.42 ± 0.2 c	0.42 ± 0.01 c
Control	10.23 ± 0.3 a	0.877 ± 0.02 a

**Table 3 molecules-27-00014-t003:** The effect of different *Cuscuta reflexa* (*Cr*) extracts on total phenolics content (TPC), total phenolic production (TPP), total flavonoids content (TFC), total flavonoids production (TFP), total polyphenolics content (TPPC), and total polyphenolics production (TPPP) in *Stevia rebaudiana*. Mean data along with standard deviation (±SD) and least significant differences were determined using Statistix software (v. 8.2; USA). Mean values with common alphabets are significantly different at *p* < 0.05.

Plant Extract	TPC mg/g-GAE-DW	TPP g/3 Flask	TFC mg/g-QE-DW	TFP g/3 Flask	TPPC g/3 Flask	TPPP g/3 Flask
*Cr* (10 mg/L)	0.06 ± 0.01 d	0.05 ± 0 c	0.01 ± 0 c	0.08 ± 0.01 c	0.07 ± 0.01 c	0.06 ± 0.01 c
*Cr* (20 mg/L)	0.1 ± 0.01 c	0.07 ± 0.01 c	0.04 ± 0 c	0.03 ± 0 c	0.14 ± 0.01 c	0.11 ± 0 b
*Cr* (30 mg/L)	0.17 ± 0.02 b	0.14 ± 0.01 ab	0.07 ± 0.01 c	0.06 ± 0.01 c	0.24 ± 0.02 b	0.19 ± 0.01 b
*Cr* (40 mg/L)	0.19 ± 0.02 b	0.12 ± 0.01 b	0.092 ± 0.02 a	0.058 ± 0.02 a	1.11 ± 0.09 a	0.07 ± 0.01 c
*Cr* (50 mg/L)	0.19 ± 0.01 b	0.14 ± 0.01 ab	0.12 ± 0.01 b	0.08 ± 0.01 c	0.32 ± 0.02 b	0.23 ± 0.01 ab
*Cr* (60 mg/L)	0.21 ± 0.02 b	0.13 ± 0.01 b	0.16 ± 0.01 b	0.09 ± 0.01 c	0.37 ± 0.03 b	0.23 ± 0.02 ab
*Cr* (70 mg/L)	0.25 ± 0.02 ab	0.15 ± 0.01 ab	0.19 ± 0.02 b	0.12 ± 0 b	0.44 ± 0.02 b	0.26 ± 0.01 ab
*Cr* (80 mg/L)	0.28 ± 0.02 a	0.19 ± 0.02 a	0.21 ± 0.01 b	0.14 ± 0 b	0.48 ± 0.01 b	0.33 ± 0.02 a
*Cr* (90 mg/L)	0.29 ± 0.01 a	0.19 ± 0.02 a	0.21 ± 0.02 b	0.14 ± 0.01 b	0.49 ± 0.03 b	0.33 ± 0.01 a
*Cr* (100 mg/L)	0.31 ± 0.03 a	0.15 ± 0.03 ab	0.22 ± 0.02 b	0.11 ± 0 b	0.53 ± 0.02 b	0.22 ± 0.01 ab
Control	0.18 ± 0.01 b	0.12 ± 0.02 b	0.11 ± 0.01 b	0.11 ± 0 b	0.27 ± 0.01 b	0.17 ± 0 b

**Table 4 molecules-27-00014-t004:** The effect of different *Cuscuta reflexa* (*Cr*) extracts on antioxidant activity in adventitious roots of *Stevia rebaudiana*. Mean values were taken from triplicated experiment. The mean values along with SD (±) with common alphabets was determined by Statistix v. 8.2 (USA) software.

Plant Extract	Antioxidant Activity (%)
*C. reflexa* (10 mg/3 flasks)	54.2 ± 0.2 g
*C. reflexa* (20 mg/3 flasks)	57.7 ± 0.7 f
*C. reflexa* (30 mg/3 flasks)	61.6 ± 0.4 e
*C. reflexa* (40 mg/3 flasks)	67.9 ± 1.0 d
*C. reflexa* (50 mg/3 flasks)	72.1 ± 0.4 c
*C. reflexa* (60 mg/3 flasks)	77.1 ± 1.3 b
*C. reflexa* (70 mg/3 flasks)	79.8 ± 0.9 ab
*C. reflexa* (80 mg/3 flasks)	81.3 ± 2.3 ab
*C. reflexa* (90 mg/3 flasks)	84.7 ± 3.1 a
*C. reflexa* (100 mg/3 flasks)	85.5 ± 3.3 a
Control w/o extract	74.4 ± 2.1 bc

## Data Availability

Data will only be available on request to the corresponding author.

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
