# Peer review of "Elicitation of Submerged Adventitious Root Cultures of Stevia rebaudiana with Cuscuta reflexa for Production of Biomass and Secondary Metabolites"

_molecules, 2021, doi:10.3390/molecules27010014_

Round 1

Reviewer 1 Report

It looks like an interesting study.  However, Results and Materials and Methods sections should be re-written to make the study understandable. Chemicals used in this study should be specified in Materials section. Methods are not clearly explained. Results and Materials and Methods parts contain many repetitions. Moreover, there is no discussion. 

Author Response

Review 1

Open Review

English language and style

( ) Extensive editing of English language and style required
( ) Moderate English changes required
(x) English language and style are fine/minor spell check required
( ) I don't feel qualified to judge about the English language and style

Yes

Can be improved

Must be improved

Not applicable

Does the introduction provide sufficient background and include all relevant references?

( )

( )

(x)

( )

Is the research design appropriate?

( )

( )

( )

( )

Are the methods adequately described?

( )

( )

( )

(x)

Are the results clearly presented?

( )

( )

( )

(x)

Are the conclusions supported by the results?

( )

( )

(x)

( )

Comments and Suggestions for Authors

It looks like an interesting study.  However, Results and Materials and Methods sections should be re-written to make the study understandable. Chemicals used in this study should be specified in Materials section. Methods are not clearly explained. Results and Materials and Methods parts contain many repetitions. Moreover, there is no discussion. 

Response: Dear Professor, thank you very much for providing us with an opportunity to revise this manuscript. You have critically checked the whole manuscript and pointed out line-by-line mistakes. We have incorporated changes suggested by you and tried our level best to enhance the quality of the manuscript. Dear Sir, you critically reviewed the manuscript and suggest us for very productive modifications, which will enhance the quality as well as data presentation. We spent day and nights to revise this manuscript very carefully. All the mistakes and errors have been removed carefully and revised the whole text according to the your comments. Hope that it will be considered for publication in this High-ranking Journal. 

We have added new references and provide sufficient background and include relevant and latest references in the introduction section. We have read the whole manuscript and tried our level best to remove grammatical and other language mistakes.

We have added new relevant references in the introduction

We have tried and re-write the methodology as well as the results to make the study understandable according to your suggestions.

We have specified all the chemicals used in this study accordingly.

We have tried to explain the methodology section clearly according to your suggestions.

We have removed the repetition and added a huge discussion accordingly.

Review 2

Open Review

English language and style

( ) Extensive editing of English language and style required
(x) Moderate English changes required
( ) English language and style are fine/minor spell check required
( ) I don't feel qualified to judge about the English language and style

Yes

Can be improved

Must be improved

Not applicable

Does the introduction provide sufficient background and include all relevant references?

( )

( )

(x)

( )

Is the research design appropriate?

(x)

( )

( )

( )

Are the methods adequately described?

(x)

( )

( )

( )

Are the results clearly presented?

(x)

( )

( )

( )

Are the conclusions supported by the results?

(x)

( )

( )

( )

Comments and Suggestions for Authors

Response: Dear Professor,

Thank you very much for your kind efforts to review our manuscript and suggested very nice and productive comments. You have studied our manuscript very carefully and it is kind of you to spend your valuable time on reviewing our manuscript. We followed your nice comments and tried to improve our manuscript according to your suggestions. We have revised the all the sections and corrected the grammatical and other language mistakes accordingly.

Authors have performed elicitation studies of adventitious root cultures of Stevia rebaudiana. 

Before publication authors should revise the manuscript based on the comments

  1. In the abstract Asteraceae  should be written in italics

Response: Dear Professor, in the abstract as well as in the whole test we write the family and genus name italic accordingly.

  1. Full form of abbreviated words should be spelled out when they are introduced the first time

Response: Dear Professor, We have followed your suggestions in the whole text.

  1. Uniformity should be maintained like ""steviol" somewhere written in capital and somewhere in small.

Response: Dear Professor, We have corrected the word “Steviol” in the whole MS accordingly.

  1. Some lines are important but do not contain references.

Response: Dear Professor, we have tried to incorporate appropriate references in the whole text.

  1. Location of the bioactive compound in plant part should be mentioned

Response: Dear Professor, we have tried to mentioned the part that synthesize bioactive compounds.

  1. Discussion should be improved and recent references should be incorporated

Response: Dear Professor, we have tried to improve the discussion by the addition of latest relevant references.

  1. Why only NAA is used

Response: Dear Professor, we have already established protocols for adventitious roots (published) in which we used NAA as the best hormone for adventitious roots development in liquid media that is why we only mentioned NAA in this study.

  1. What is AS, AB, CF, DF, etc

Response: Dear Professor, we have explain these abbreviation in the methodology section accordingly.

  1. Standard graphs should be added

Response: Dear Professor, we have added standard graph accordingly.

  1. Suggestion to add these references for adventitious root culture and elicitation studies

Response: Dear Professor, we have incorporated these references in the revised MS accordingly.

Roy, A., & Bharadvaja, N. (2019). Establishment of root suspension culture of Plumbago zeylanica and enhanced production of plumbagin. Industrial Crops and Products, 137, 419-427.

Fan, M. Z., An, X. L., Cui, X. H., Jiang, X. L., Piao, X. C., Jin, M. Y., & Lian, M. L. (2021). Production of eurycomanone and polysaccharides through adventitious root culture of Eurycoma longifolia in a bioreactor. Biochemical Engineering Journal, 171, 108013.

Krishnan, M. L., Roy, A., & Bharadvaja, N. (2019). Elicitation effect on the production of asiaticoside and asiatic acid in shoot, callus, and cell suspension culture of Centella asiatica. Journal of Applied Pharmaceutical Science, 9(06), 067-074.

Kundu, K., Roy, A., Saxena, G., Kumar, L., & Bharadvaja, N. (2016). Effect of different carbon sources and elicitors on shoot multiplication in accessions of Centella asiatica. Med Aromat Plants, 5(251), 2167-0412.

Review 3

Open Review

English language and style

( ) Extensive editing of English language and style required
( ) Moderate English changes required
(x) English language and style are fine/minor spell check required
( ) I don't feel qualified to judge about the English language and style

Yes

Can be improved

Must be improved

Not applicable

Does the introduction provide sufficient background and include all relevant references?

( )

(x)

( )

( )

Is the research design appropriate?

(x)

( )

( )

( )

Are the methods adequately described?

( )

(x)

( )

( )

Are the results clearly presented?

(x)

( )

( )

( )

Are the conclusions supported by the results?

( )

(x)

( )

( )

Comments and Suggestions for Authors

I found the manuscript " Elicitation of submerged Adventitious Root Cultures of Stevia 2 rebaudiana with Cuscuta reflexa for Production of Biomass and 3 Secondary Metabolites " by Nisar et al. interesting, which describes effect of Cuscuta reflexa extract as biotic elicitor on biomass accumulation and production of antioxidant secondary metabolite in submerged adventitious root culture of Stevia rebaudiana. The topic is of current interest and suited for the journal; anyways, some modifications of the submitted paper are recommended before publication.

Response: Dear Professor,

Thank you very much for your valuable comments. Your comments really impressed me. We are happy for the chance to eliminate these weaknesses in our manuscript. Dear Sir, we specially focused on your comments and it will really modify the revised manuscript. However, we have tried our best to incorporate all of changes/modifications suggested by you. We have revised the introduction, methodology section and conclusion accordingly. Hope that it will be satisfactory this time.

Comments and remarks:

Abstract:

Abstract needed to be revised.

Response: Dear Professor, we have revised the whole ABSTRACT accordingly.

Introduction

Kindly mention the geographical distribution of Stevia rebaudiana plants.

Response: Dear Professor, we have incorporated the geographical location of Stevia plants in the revised MS accordingly.

Explain the phytochemicals which are present in this plant with updated references.

Response: Dear Professor, we have added the main phytochemicals produce by the leaves of stevia in the introduction section.

Also, describe the medicinal value of this plant.

Response: Dear Professor, We have describe the medicinal importance of Stevia as well as Cuscuta in the introduction section accordingly.

Results:

The quality of figure 1 should be enhanced

Response: Dear Professor, we tried our level best to enhance the quality of figure 1 accordingly.

Figure 1 must be addressed appropriately, especially the labeling like A, B ..etc.

Response: Dear Professor, we have correctly addressed and explain the various cultures (A, B,…….etc.) in the figure caption as well as in the results section as figure 1a, figure 1b, figure 1c……..etc. according to your suggestions.

Some grammatical and typo errors are found.

 Authors can take help English native speakers for English editing.

Response: Dear Professor, we tried our level best to modify the language of the manuscript and removed grammatical and typo errors in the whole text.

Add some new references

Response: Dear Professor, we have added maximum number of latest and relevant references in all the sections of the revised manuscript.

Submitted for your kind consideration

Professor, Dr. Bilal Haider Abbasi,

bhabbasi@qau.edu.pk

Reviewer 2 Report

Authors have performed elicitation studies of adventitious root cultures of Stevia rebaudiana. 

Before publication authors should revise the manuscript based on the comments

  1. In the abstract Asteraceae  should be written in italics
  2. Full form of abbreviated words should be spelled out when they are introduced the first time
  3. Uniformity should be maintained like ""steviol" somewhere written in capital and somewhere in small.
  4. Some lines are important but do not contain references.
  5. Location of the bioactive compound in plant part should be mentioned
  6. Discussion should be improved and recent references should be incorporated
  7. Why only NAA is used
  8. What is AS, AB, CF, DF, etc
  9. Standard graphs should be added
  10. Suggestion to add these references for adventitious root culture and elicitation studies

Roy, A., & Bharadvaja, N. (2019). Establishment of root suspension culture of Plumbago zeylanica and enhanced production of plumbagin. Industrial Crops and Products, 137, 419-427.

Fan, M. Z., An, X. L., Cui, X. H., Jiang, X. L., Piao, X. C., Jin, M. Y., & Lian, M. L. (2021). Production of eurycomanone and polysaccharides through adventitious root culture of Eurycoma longifolia in a bioreactor. Biochemical Engineering Journal, 171, 108013.

Krishnan, M. L., Roy, A., & Bharadvaja, N. (2019). Elicitation effect on the production of asiaticoside and asiatic acid in shoot, callus, and cell suspension culture of Centella asiatica. Journal of Applied Pharmaceutical Science, 9(06), 067-074.

Kundu, K., Roy, A., Saxena, G., Kumar, L., & Bharadvaja, N. (2016). Effect of different carbon sources and elicitors on shoot multiplication in accessions of Centella asiatica. Med Aromat Plants, 5(251), 2167-0412.

Author Response

(The authors gave the same response as above.)

Reviewer 3 Report

I found the manuscript " Elicitation of submerged Adventitious Root Cultures of Stevia 2 rebaudiana with Cuscuta reflexa for Production of Biomass and 3 Secondary Metabolites " by Nisar et al. interesting, which describes effect of Cuscuta reflexa extract as biotic elicitor on biomass accumulation and production of antioxidant secondary metabolite in submerged adventitious root culture of Stevia rebaudiana. The topic is of current interest and suited for the journal; anyways, some modifications of the submitted paper are recommended before publication.

Comments and remarks:

Abstract:

Abstract needed to be revised.

Introduction

Kindly mention the geographical distribution of Stevia rebaudiana plants.

Explain the phytochemicals which are present in this plant with updated references.

Also, describe the medicinal value of this plant.

Results:

The quality of figure 1 should be enhanced

Figure 1 must be addressed appropriately, especially the labeling like A, B ..etc.

Some grammatical and typo errors are found.

 Authors can take help English native speakers for English editing.

Add some new references

Author Response

(The authors gave the same response as above.)

Round 2

Reviewer 2 Report

The authors have incorporated suggested changes. Therefore manuscript can be accepted.